# Clicking Azides and Alkynes with Poly(pyrazolyl)borate-Copper(I) Catalysts: An Experimental and Computational Study

**Lahoucine Bahsis** [1], **Hicham Ben El Ayouchia** [1,*], **Hafid Anane** [1], **Carmen Ramirez de Arellano** [2], **Abdeslem Bentama** [3], **El Mestafa El Hadrami** [3,*], **Miguel Julve** [4], **Luis R. Domingo** [2] **and Salah-Eddine Stiriba** [1,4,*]

[1] Laboratoire de Chimie Analytique et Moléculaire/LCAM, Faculté Polydisciplinaire de Safi, Université Cadi Ayyad, Safi 46030, Morocco

[2] Departamento de Química Orgánica, Universidad de Valencia, Avda. Dr. Moliner 50, Burjassot, 46100 Valencia, Spain

[3] Laboratoire de Chimie Organique Appliquée (LCAO), Faculté des Sciences et Techniques, Université Sidi Mohamed Ben Abdellah, BP 2202 Fès, Morocco.

[4] Instituto de Ciencia Molecular/ICMol, Universidad de Valencia, C/Catedrático José Beltrán 2, Paterna, 46980 Valencia, Spain

[*] Correspondence: belayou@gmail.com (H.B.E.A.); elmestafa.elhadrami@usmba.ac.ma (E.M.E.H.); stiriba@uv.es (S.-E.S.); Tel.: +34-96-354-44-45 (S.-E.S.)

**Abstract:** The synthesis of 1,4-disubstitued-1,2,3-triazoles under a copper(I)-catalyzed azide–alkyne cycloaddition (CuAAC) regime was accomplished in high yields and a regioselective manner by using two homoscorpionate poly(pyrazolyl)borate anions: tris(pyrazolyl)hydroborate ($HB(pz)_3^-$) and bis(pyrazolyl)hydroborate ($H_2B(pz)_2^-$), which stabilized in situ the catalytically active copper (I) center. The [3+2] cycloaddition (32CA) reactions took place under strict click conditions, including room temperature and a mixture of environmentally benign solvents such as water/ethanol in a 1:1 (v/v) ratio. These click chemistry conditions were applied to form complex 1,2,3-triazoles-containing sugar moieties, which are potentially relevant from a biological point of view. Computational modeling carried out by DFT methodologies at the B3LYP/6-31G(d) level showed that the coordination of poly(pyrazolyl)borate-copper(I) to alkyne groups produced relevant changes in terms of generating a high polar copper(I)-acetylide intermediates. The analysis of the global and local reactivity indices explains correctly the role of poly(pyrazolyl)borate ligands in the stabilization and activation of the copper(I) catalyst in the studied 32CA reactions.

**Keywords:** click chemistry; azides; alkynes; copper(I); poly(pyrazolyl)borate; 1,2,3-triazole; DFT calculations

## 1. Introduction

During the previous decade, click chemistry emerged as a powerful concept for the preparation of a diversity of structurally complex chemical and biological systems under challenging conditions, mainly via the efficient formation of carbon–heteroatom bonds. The most fashionable and premier click chemistry reaction that respects the clickable chemistry criteria is the copper(I)-catalyzed [3+2] cycloaddition (32CA) reaction of azide with alkyne (CuAAC) that leads to regioselective 1,4-disubstitued-1,2,3-triazole linkages viewed nowadays as non-classical heterocyclic bioisosteres [1]. The mixture of copper(II) sulphate pentahydrate and sodium ascorbate, which has been commonly used as a precatalyst system for CuAAC reactions, generates catalytically active copper(I) species in

the reaction media [2]. Later, it was found that polydentate nitrogen ligands not only stabilize copper(I) ions [3] but also accelerate the catalytic process [4], permitting, then, the direct employment of copper(I) ions as catalysts in CuAAC reactions. Nevertheless, the direct use of copper(I) presents several problems under aerobic experimental conditions because of its thermodynamic instability and easy oxidation to copper(II). The search for stabilizing ligands that would keep the less Lewis acid copper(I) catalytically active is crucial and in high demand in developing CuAAC as a common synthetic methodology of choice [5]. Indeed, several nitrogen-containing polydentate ligands have been used, keeping in mind that their tetradentate binding ability will completely wrap and protect the copper(I) ionic center against potential destabilizing reactants present in the medium. Some of the interesting ligands include the dipicolinate [6], tris-(benzyltriazolylmethyl)amine (TBTA) [7], the crowded tetradentatetris(2-dioctadecylaminoethyl)amine (C186tren) [8], tris(triazolyl)methanol (TBTM) [9], and tris(pyrazolyl)methane (Tpm)—molecules which are found to be effective in the formation of *N*-sulfonyl-1,2,3-triazoles from *N*-sulfonylazides and alkynes (Figure 1) [10]. The well-known homoscorpionate ligands based on tris(pyrazolyl)hydroborate and bis(pyrazolyl)dihydroborate anions tris(pyrazolyl)hydroborate (HB(pz)$_3^-$) and bis(pyrazolyl)hydroborate (H$_2$B(pz)$_2^-$) are among the powerful and familiar polydentate nitrogen-containing species that have demonstrated to be versatile ligands useful for the preparation of copper(I) complexes [11–17] (Figure 1). The principal feature in all poly(pyrazolyl)borate complexes is the formation of the six-membered ring that stabilizes the copper(I) ion [18]. These complexes have been used as bioinorganic mimicking systems and also in several catalytic organic processes, such as C–H insertion and cyclopropanation reactions of diazo compounds [19]. However, their catalytic potential activity has been scarcely explored in CuAAC, with the employed tris(pyrazolyl)borate (Tpx, X = Ph, Br, or Me) being inactive in catalyzing the synthesis of *N*-sulfonyl-1,2,3-triazoles in halogenated solvents [10]. Herein, we report on the use of the homoscropionate poly(pyrazolyl)borate anions HB(pz)$_3^-$ and H$_2$B(pz)$_2^-$ as stabilizing ligands of copper(I) in situ and on their application in ligating a variety of azides and alkynes under very strict click chemistry conditions. The role of the poly(pyrazolyl)borate-copper(I) in enhancing the catalytic activity was addressed by computational modeling by means of DFT methodology.

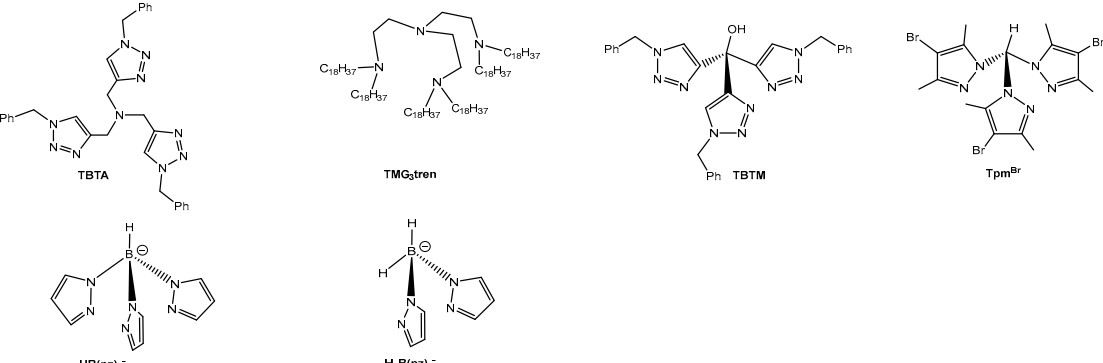

**Figure 1.** Structure of polydentate nitrogen ligands previously studied in copper(I)-catalyzed azide–alkyne cycloaddition (CuAAC) and the poly(pyrazolyl)borates tris(pyrazolyl)hydroborate (HB(pz)$_3^-$) and bis(pyrazolyl)hydroborate (H$_2$B(pz)$_2^-$) explored in this work.

## 2. Results and Discussion

### 2.1. Poly(pyrazolyl)borate Copper(I)-Catalyzed Alkyne–Azide 32CA Reaction

The presence of two structural features such the triazole groups and the tripodal structure in the poly(pyrazolyl)borate anions (HB(pz)$_3^-$ and HB(pz)$_2^-$) enabled them to coordinate to a copper(I) ion with different geometries. The calculated optimized structures of [Cu{HB(pz)$_3$}L] (**1a**) and [Cu{H$_2$B(pz)$_2$}L] (**1b**) with the atom numbering are shown in Figure 2. Bond distances and angles of the complexes **1a** and **1b** are summarized in Table 1. Each copper(I) ion in **1a** was four-coordinate by

the hydrotris(pyrazolyl)borate anion, and behaved as a tridentate ligand (3N of pyrazolyl) towards the $d^{10}$ copper(I) center and by one oxygen atom of one water molecule, thereby completing a somewhat distorted tetrahedral arrangement and also achieving the 18-outer-electron (noble gas) configuration of this atom. Each copper(I) ion in **1b** was trigonally coordinated by two nitrogen atoms of pyrazolyl (bidentate ligand) and the oxygen atom of one water molecule. The Cu–O distance in **1b** (2.04 Å) was marginally shorter than that in **1a** (2.09Å). The mean Cu–$N_{pz}$ distance for the tridentate ligand was 2.097 Å, a value slightly longer than that in the bidentate coordination 1.196 Å.

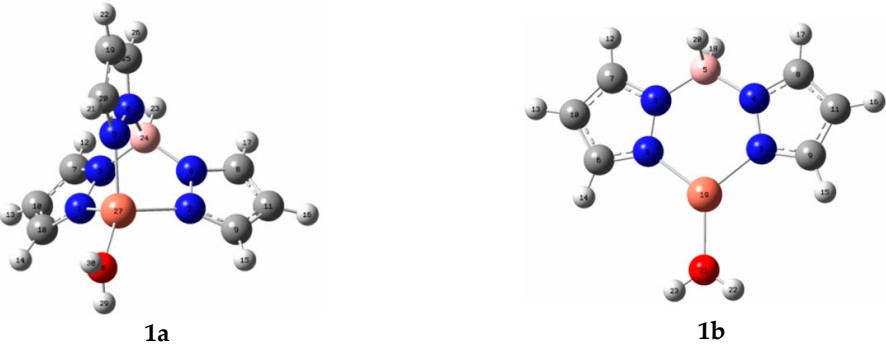

**1a**                **1b**

**Figure 2.** Optimized structures of the complexes **1a** and **1b**.

The intramolecular N–Cu–N angles were all remarkably less than the ideal tetrahedral angle of 109.5°, due to a combination of the constraints within the hydro tris(pyrazolyl)borate ligand and the requirements for normal copper–nitrogen bond distances. The individual values in **1a** ranged from 93.14 [N(1)–Cu–N(2)] to 93.98° [N(1)–Cu–N(3)]. This automatically influenced the N(pyrazoly1)–Cu–O(carbonyl) angles, all of them being noticeably greater than the ideal tetrahedral angle and ranging from 119.44 to 128.41°. These findings were in good agreement with the X-ray crystallography recorded by Churchill et al. [20]. The value of 104.98° for N(1)–Cu–N(2) in **1b** was notably smaller than the ideal trigonal angle of 120°, a feature that affects those of the N(pyrazoly1)–Cu–O(carbonyl) angles, all of them being slightly higher than the ideal trigonal angle (values covering the range of 127.42 to 127.59°).

**Table 1.** Selected theoretical parameters of the geometry of **1a** and **1b** and their comparison with the experimental parameters [20,21].

|  | 1a | 1b | Exp [a] | Exp [b] |
|---|---|---|---|---|
| **Bond Lengths (Å)** | | | | |
| Cu–O | 2.090 | 2.008 | - | 1.918 |
| Cu–N1 | 2.109 | 1.966 | 2.039 | 2.004 |
| Cu–N5 | 2.075 | 1.966 | 2.059 | 2.061 |
| Cu–N2 | 2.109 | - | 2.039 | 2.094 |
| N1–N3 | 1.390 | 1.386 | 1.378 | |
| N2–N4 | 1.390 | 1.386 | 1.353 | |
| N5–N6 | 1.390 | - | 1.350 | |
| N3–B | 1.553 | 1.578 | 1.526 | - |
| N4–B | 1.555 | 1.578 | 1.558 | - |
| N6–B | 1.553 | - | 1.540 | - |
| **Bond Angles (°)** | | | | |
| O–Cu–N1 | 119.447 | 127.425 | - | 131.9 |
| O–Cu–N2 | 128.412 | 127.592 | - | 114.5 |
| O–Cu–N5 | 119.596 | - | - | 121.7 |
| N1–Cu–N2 | 93.974 | 104.981 | 90.4 | 92.7 |
| N1–Cu–N5 | 93.149 | | 92.4 | 92.1 |

| N2–Cu–N5 | 93.982 | - | 90.8 | 95.4 |

[a] Experimental values from [20]; [b] Experimental values from [21].

The catalytic activity of the poly(pyrazolyl)borate-copper(I) complex **1a** and **1b,** generated in situ by reacting CuCl with the corresponding ligand in a water/ethanol solvent mixture, was investigated in the 32CA reaction between benzyl azide and phenyl acetylene. This 32CA reaction was chosen as reaction model, and its behavior was studied under a variety of conditions and monitored using thin layer chromatography (TLC) and $^1$H NMR spectroscopy (Table 2).

During our optimized conditions, the 32CA reaction between benzyl azide and phenyl acetylene in the presence of 5 mol% of CuCl without a ligand and in a water/ethanol (1:1 *v/v*) solvent mixture produced the 1,4-disubstituted-1,2,3-triazole in a yield ca. 50% after stirring for 48 h at room temperature. By repeating the experiment in the presence of the ligand [$H_2B(pz)_2^-$ or $HB(pz)_3^-$], the 32CA reaction was completed within 24 h and the best performances in terms of the conversion favoring the 1,4-disubstituted triazoles were 84 and 95% (see Table 2).

**Table 2.** Conditions screening for the [3+2] cycloaddition (32CA) reaction of benzyl azide and phenylacetylene.

| Entry | Ligand | Solvent | Time (h) | Yield (%) |
|---|---|---|---|---|
| 1 | - | Toluene | 48 | - |
| 2 | - | H₂O/EtOH | 48 | 50 |
| 3 | HB(Pz)₃⁻ | CH₃CN | 24 | 89 |
| 4 | H₂B(Pz)₂⁻ | CH₃CN | 24 | 78 |
| 5 | HB(Pz)₃⁻ | CH₃OH | 24 | 92 |
| 6 | H₂B(Pz)₂⁻ | CH₃OH | 24 | 83 |
| 7 | HB(Pz)₃⁻ | EtOH | 24 | 91 |
| 8 | H₂B(Pz)₂⁻ | EtOH | 24 | 80 |
| 9 | HB(Pz)₃⁻ | H₂O | 24 | 92 |
| 10 | H₂B(Pz)₂⁻ | H₂O | 24 | 83 |
| 11 | HB(Pz)₃⁻ | H₂O/EtOH | 24 | 95 |
| 12 | H₂B(Pz)₂⁻ | H₂O/EtOH | 24 | 84 |

The poly(pyrazolyl)borate-copper(I) complexes were found to be highly catalytically active under very mild conditions. Indeed, various solvents were examined (Table 2), pointing out that the solvent plays a significant role concerning the obtained yield. Water and ethanol clearly stand out as the solvents of choice for high yields, selectivity, cheapness, and an environmentally benign nature. The reactions between benzyl azide and phenylacetylene together with 5 mol% of CuCl and 5 mol% of poly(pyrazolyl)borate anion in a water/ethanol (1:1 v/v) mixture yielded the 1,4-disubstituted triazole in 95% yield after stirring for 24 h at room temperature.

With the optimized conditions (Table 2, entries 11–12) at hand, the generality and versatility of this method was checked with various structurally-diverse terminal alkynes such as *para*-substituted phenylacetylenes, with both electron-donating and electron-withdrawing substituents, and sugar-alkynes with various azides, such as alkyl azides, benzyl azides, and sugar azides. The 32CA reactions were finished in a reasonable time in all cases and the corresponding 1,2,3-triazoles were isolated in good to high yields (Table 3, See Supplementary Materials for more characterization details).

**Table 3.** 32CA reactions of azides and alkynes catalyzed by poly(pyrazolyl)borate-copper(I).

| Entry | Alkyne | Azide | Product | Ligand | Yield [a] (%) | TON [b] |
|---|---|---|---|---|---|---|
| 1 | | | **3a** | HB(pz)₃⁻ | 92 | 18.4 |
| | | | | H₂B(pz)₂⁻ | 84 | 16.8 |
| 2 | | | **3b** | HB(pz)₃⁻ | 93 | 18.4 |
| | | | | H₂B(pz)₂⁻ | 89 | 17.8 |
| 3 | | | **3c** | HB(pz)₃⁻ | 89 | 17.8 |
| | | | | H₂B(pz)₂⁻ | 87 | 17.4 |
| 4 | | | **3d** | HB(pz)₃⁻ | 76 | 15.2 |
| | | | | H₂B(pz)₂⁻ | 70 | 14 |
| 5 | | | **3e** | HB(pz)₃⁻ | 92 | 18.4 |
| | | | | H₂B(pz)₂⁻ | 88 | 17.6 |
| 6 | | | **3f** | HB(pz)₃⁻ | 89 | 17.8 |
| | | | | H₂B(pz)₂⁻ | 86 | 17.2 |
| 7 | | | **3g** | HB(pz)₃⁻ | 88 | 17.6 |
| | | | | H₂B(pz)₂⁻ | 82 | 16.4 |
| 8 | | | **3h** | HB(pz)₃⁻ | 73 | 14.6 |
| | | | | H₂B(pz)₂⁻ | 70 | 14 |
| 9 | | | **3i** | HB(pz)₃⁻ | 83 | 16.6 |
| | | | | H₂B(pz)₂⁻ | 80 | 16 |
| 10 | | | **3j** | HB(pz)₃⁻ | 89 | 17.8 |
| | | | | H₂B(pz)₂⁻ | 85 | 17 |
| 11 | | | **3k** | HB(Pz)₃⁻ | 80 | 16 |
| | | | | H₂B(Pz)₂⁻ | 78 | 78 |
| 12 | | N₃–(CH₂)₁₂–N₃ | **3l** | HB(Pz)₃⁻ | 90 | 18 |
| | | | | H₂B(Pz)₂⁻ | 77 | 15.4 |

| | | | | | | |
|---|---|---|---|---|---|---|
| 13 | *(carbohydrate propargyl ether structure)* | N₃–(CH₂)₁₂–N₃ | **3m** | HB(pz)₃⁻ | 80 | 16 |
| | | | | H₂B(pz)₂⁻ | 78 | 15.6 |
| 14 | *(carbohydrate propargyl ether structure)* | N₃–(CH₂)₁₂–N₃ | **3n** | HB(pz)₃⁻ | 85 | 17 |

[a] Isolated yield (%); [b] TON = turnovers number (moles substrate/moles of catalyst).

The regioselectivity of the 32CA reactions of azides with alkynes was confirmed by the X-ray analysis on a single crystal of compound **3h**. Its molecular structure was unequivocally determined to be 4-(1-benzyl-1H-1,2,3-triazol-4-yl)aniline, and it corresponds to the expected 1,4-regioisomer of the 1,2,3-tryazol heterocyclic product from the 32CA reaction of *p*-aminophenyl acetylene and benzylazide (Figure 3). The carbon–carbon and carbon–nitrogen bond lengths in the triazole ring agreed with the expected values for other described triazole systems (Figure 3). The mean plane of the triazole ring formed dihedral angles of 78.12(6) and 11.30(9)° with the phenyl ring mean planes corresponding to C(11) and C(41), respectively. The amino group, N(4), and the phenyl ring, C(41)–C(46), were not in the same plane, but they formed a dihedral angle of 38.5(1)°. The most relevant interactions found in the crystal packing were those corresponding to a N–H···N hydrogen bond [N(4)–H(1)···N(3)ᴵ with N(4)···N(3)ᴵ = 3.280(2) Å, H(1)···N(3)ᴵ = 2.371(18) Å, and N(4)–H(1)···N(3)ᴵ = 169.2(17)°; symmetry code (i) = −x + 5/2, −y + 1, z + 1/2] and a N–H···π-type interaction with the C(11)–C(16) ring [H(2)···C(11)ⁱⁱ = 3.006(21) Å, H(2)···C(12)ⁱⁱ = 2.860(19) Å, H(2)···C(13)ⁱⁱ = 2.727(19) Å, H(2)···C(14)ⁱⁱ = 2.742(20) Å, H(2)···C(15)ⁱⁱ = 2.877(21) Å, and H(2)···C(16)ⁱⁱ = 2.997(21) Å; symmetry code (ii) = x + 1/2, −y + 1/2, −z + 2] (Figure 3, right).

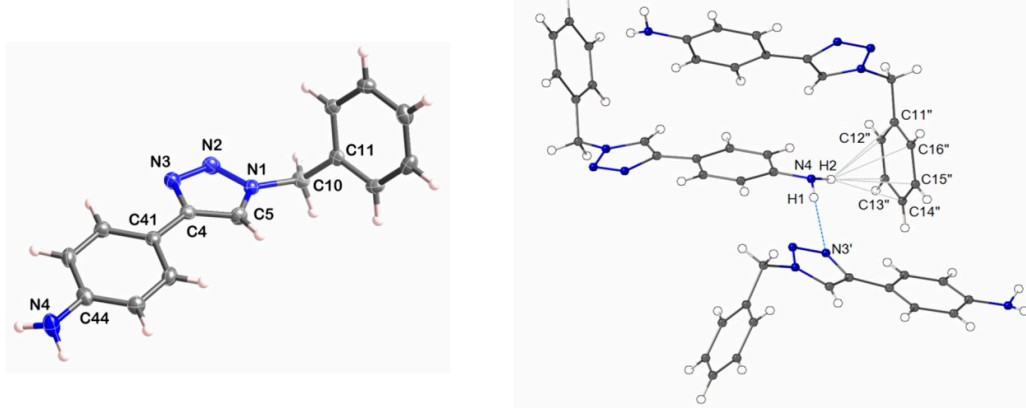

**Figure 3.** (**Left**) Ellipsoid plot (50% probability level) for **3h** with the atom numbering scheme. (**Right**) Packing view showing the interactions involving the NH₂ group. Selected bond lengths and angles (Å°): N(1)–N(2), 1.3409(19); N(1)–C(5), 1.342(2); N(1)–C(10), 1.468(2); N(2)–N(3), 1.3211(19); N(3)–C(4), 1.367(2); C(4)–C(5), 1.373(2); N(2)–N(1)–C(5), 111.07(14); N(3)–N(2)–N(1), 106.95(14); N(2)–N(3)–C(4), 109.14(13); N(3)–C(4)–C(5), 107.43(14); and N(1)–C(5)–C(4), 105.41(15).

**Table 4.** Catalytic performance of poly(pyrazolyl)borate-copper(I) systems with other literature reports in CuAAC reactions.

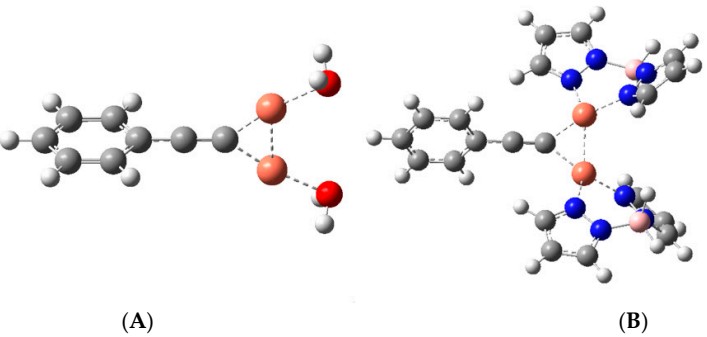

| Entry [a] | Cu Loading (mol %) | Solvent | T (°C) | Time (h) | Yield (%) | References |
|---|---|---|---|---|---|---|
| Cu(I)-HB(pz)$_3^-$ | 5 | EtOH/H$_2$O | 25 | 24 | 92 | This work |
| Cu(I)-H$_2$B(pz)$_2^-$ | 5 | EtOH/H$_2$O | 25 | 24 | 84 | This work |
| Cu(II)-tren | 0.2 | *n*-Octane | 25 | 24 | 84 | [22] |
| Cu(I)-tren | 0.05 | Toluene | 60 | 24 | 86 | [7] |
| Cu(I)-TBTM | 0.5 | H$_2$O | 25 | 4 | 94 | [8] |
| Cu(I)-TBTA | 1 | *t*-BuOH/H$_2$O | 25 | 24 | 84 | [19] |
| C$_3$H$_7$COOCu(PPh$_3$)$_2$ | 0.15 | Dichloromethane | 28 | 3 | 99 | [23] |

[a] Abbreviations: HB(pz)$_3^-$ = tris(pyrazolyl)hydroborate; H$_2$B(pz)$_2^-$ = bis(pyrazolyl)hydroborate; tren = tris(2-dioctadecylaminoethyl)amine; TBTM = tris(triazolyl)methanol; TBTA = tris(benzyltriazolylmethyl)amine.

The click of 1,4-disubstituted-1,2,3-triazoles under CuAAC reactions has already been performed with a number of copper(I)-containing poly(nitrogen) coordinating ligands [22, 7,8,19]. The reaction conditions and their catalytic performance are summarized in Table 4. It can be noted that apart from the Cu(I)-TBTM and Cu(I)-TBTA catalysts, the other catalytic systems required the use of organic solvents as a reaction medium. Our catalytic systems showed higher activity in CuAAC by using a mixture of environmentally-benign solvents, such as ethanol and water, in a similar performance to those previously reported, in particular to the well-studied Cu(I)-TBTM system.

## 2.2. Global and Local Conceptual DFT (CDFT) Reactivity Index Analysis

Global reactivity indices defined within the conceptual DFT [24,25] are powerful tools to understand the reactivity in cycloaddition reactions. Since the global electrophilicity and nucleophilicity scales are given at the B3LYP/6-31G(d) level, reactants were optimized at the same computational level. The global reactivity indexes, such as electronic chemical potential (μ), chemical hardness (η), global electrophilicity (ω), and global nucleophilicity (N), for phenylazide and phenyl acetylene, the simplest dinuclear copper-acetylides and dinuclear copper-acetylides (Figure 4), are listed in Table 5.

(A)          (B)

**Figure 4.** Structure of the simple dinuclear copper(I)-acetylide species (**A**) and dinuclear bis(pyrazolyl)borato]copper(I)-acetylide (**B**).

**Table 5.** Chemical potential (μ), chemical hardness (η), global electrophilicity (ω), and global nucleophilicity (*N*) of methylazide, methyl acetylene, and dinuclear copper-acetylide intermediates.

| Species | μ (eV) | η (eV) | ω (eV) | *N* (eV) |
|---|---|---|---|---|
| Dinuclear Cu(I)-acetylide | 1.68 | 1.09 | 1.29 | 10.25 |
| Phenylazide | −3.62 | 5.17 | 1.27 | 2.92 |
| Phenyl acetylene | −3.53 | 5.51 | 1.13 | 2.83 |
| Dinuclear Cu(I)-acetylide | 0.38 | 3.47 | 0.02 | 7.76 |

As shown in Table 5, the electronic chemical potential of phenyl acetylene, μ = −3.53 eV, was slightly larger than that of the phenylazide, μ = −3.62 eV, indicating that no global electron transfer (GEDT) [26] will occur along the corresponding 32CA reaction. However, the electronic chemical potential of μ = 1.68 (**A**) and 0.38 (**B**) eV (Figure 4A,B), were clearly higher than that of phenylazide, which shows that along a copper(I)-catalyzed 32CA reaction, the GEDT fluxes from these complexes to phenylazide.

The globalelectrophilicity and nucleophilicity indices of phenylazide were ω = 1.27 eV and *N* = 2.92 eV, respectively, classifying it as a strong electrophile and at the borderline of being classified as a strong nucleophile within the electrophilicity [27] and nucleophilicity [28] scales. On the other hand, phenylacetylene had global electrophilicity ω and nucleophilicity *N* indexes of 1.13 eV and 2.83 eV, respectively—values that allow for the classification of this species as a strong electrophile and at the borderline of being classified as a strong nucleophile. The similar electrophilic and nucleophilic character of both phenylazide and phenylacetylene show that the corresponding 32CA reaction will have a non-polar character, in clear accord with the analysis of the electronic chemical potential of these compounds.

Coordination of the copper(I) ion to the terminal carbon of alkyne producing the simplest dinuclear Cu(I)-acetylide (Figure 4A) increased the value of the ω index of the corresponding complex to 1.29 eV, allowing the classification of this species as a strong electrophile, but, remarkably, its nucleophilicity *N* index was more dramatically increased to 10.25 eV (Table 5). Moreover, the coordination of the stabilized copper(I) by dihydrobis(pyrazolyl)borate to the terminal carbon of alkyne forming a highly activated acetylene reagent [dinuclear Cu(I)-acetylide] (Figure 4B) decreased markedly in its ω index to 0.02 eV, classifying it as a marginal electrophile, and increased remarkably its nucleophilicity *N* index to 7.76 eV, classifying it as a strong nucleophile.

This high *N* index indicated that complex (B) will act as a very strong nucleophile in the 32CA reactions with a large polar character, hence making more advantageous the use of poly(pyrazolyl)borate ligands in such copper(I)-catalyzed azide–alkyne 32CA reactions.

By approaching an electrophile–nucleophile pair, the most favorable reactive channel was that associated with by the initial two-center interaction between the greatest electrophilic center of the electrophile unit and the greatest nucleophilic one of the nucleophile entity. Recently, Domingo et al. proposed the electrophilic $P_k^+$ and nucleophilic $P_k^-$ Parr functions [29] derived from the excess of spin electron-density reached via the GEDT process from the nucleophile to the electrophile as a powerful tool for the study of the local reactivity in polar processes. Thus, the electrophilic $P_k^+$ Parr functions of phenylazide and the nucleophilic $P_k^-$ Parr functions of the complexes (A) and (B) were calculated, respectively. The corresponding values are summarized in Table 6.

The examination of the electrophilic $P_k^+$ Parr functions of phenylazide showed that the non-substituted N1 nitrogen, with $P_k^+$ = 0.29, is the greatest electrophilic center of this molecule (see Table 6). Note that the N3 nitrogen presented a low electrophilic deactivation, $P_k^+$ = −0.09. On the other hand, the examination of the nucleophilic $P_k^-$ Parr functions of the complexes (A) and (B) indicated that the phenylsubstituted C4 carbon is the greatest nucleophilic center of the acetylide

framework, with $P_k^-$ = 0.28. Note that the non-substituted carbon was nucleophilicity deactivated. In these complexes, the two copper(I) ions were also nucleophilic activated centers, but they did not participate in the 32CA reaction.

Consequently, the examination of the electrophilic $P_k^+$ and the nucleophilic $P_k^-$ Parr functions showed that along a polar 32CA reaction, the more favorable regioisomeric reaction path will be that associated with the first formation of the C4–N1 single bond involving the non-substituted nitrogen atom of the azide and the substituted carbon atom of the acetylene.

**Table 6.** Electrophilic $P_k^+$ Parr functions of the phenylazide and nucleophilic $P_k^-$ Parr functions of the complexes (**A**) and (**B**).

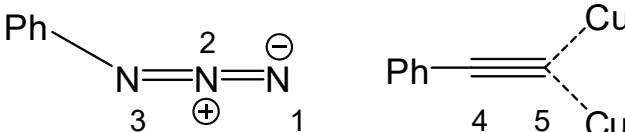

| | Number of the Atom | $P_k^+$ | $P_k^-$ |
|---|---|---|---|
| | N1 | 0.29 | |
| Phenylazide | N2 | 0.18 | |
| | N3 | −0.09 | |
| | C4 | | 0.28 |
| Simplest dinuclear Cu(I)-acetylide (**A**) | C5 | | −0.07 |
| | Cu | | 0.25 |
| Dinuclear Cu(I)-acetylide stabilized by bis(pyrazolyl)borate (**B**) | C4 | | 0.28 |
| | C5 | | −0.12 |
| | Cu | | 0.30 |

The next step in the mechanism accounting for the formation of 1,2,3-triazoles in the presence of the poly(pyrazolyl)borate-copper(I) catalyst consisted of a nucleophilic attack to the organoazide at the N3 atom by the C4 atom of the Cu(I)-acetylide forming the first covalent C–N bond and resulting into the intermediate **4** [30,31]. Because of its ring contraction, compound **4** led to the formation of the triazolyl-copper **5**. The last step corresponded to a fast protonation of the copper triazolide, triggering thereby the final triazole product (Figure 5) [32,33].

**Figure 5.** The proposed mechanism for the formation of 1,2,3-triazole catalyzed by the poly(pyrazolyl)borate-copper(I) catalyst.

## 3. Materials and Methods

### 3.1. Reagents and Physical Measurements

All chemicals were purchased from commercial sources and used as received. CuCl and the potassium salts KHB(pz)$_3$ and KH$_2$B(pz)$_2$ were purchased from Sigma-Aldrich (St. Louis, MO, USA). All the reactions were carried out in the open air. The reaction mixtures were monitored by TLC using commercial glass-backed thin layer chromatography (TLC) plates (Merck Kieselgel 60 F254). The plates were observed under UV-light at 254 nm. NMR analyses were carried out on a Brucker (Valencia, Spain) AC-400 MHz (400 and 100 MHz for proton and carbon, respectively) spectrometer by using deuterated chloroform as a solvent. The chemical shifts (δ) were expressed in ppm. The high resolution mass spectra (HRMS) were recorded in the EI (70 eV) or FAB mode at the mass spectrometry service of the University of Valencia. FT-IR spectra (400–4000 cm$^{-1}$ range) were recorded on a Nicolet 5700 FT-IR spectrometer. Melting points were determined using a Stuart melting point apparatus SMP3, employing the capillary tubes.

### 3.2. Computational Methods

All stationary points were optimized using the B3LYP functional [34,35], together with the 6-31G(d) and LANL2DZ basis sets [36]. The gas phase calculations were performed using a G09 package [37]. The catalytic role of the Cu(I) ion was studied analyzing the reactivity indices defined within the conceptual DFT (CDFT) [24,25]. The electrophilicity ω index [38], was given by the following expression $\omega = (\mu^2/2\eta)$, where μ is the electronic chemical potential and η, the chemical hardness. Both quantities may be approached in terms of the one-electron energies of the frontier molecular orbitals HOMO and LUMO, ($\varepsilon_H$ and $\varepsilon_L$) as $\mu \approx (\varepsilon_H + \varepsilon_L)/2$ and $\eta = (\varepsilon_L - \varepsilon_H)$, respectively [39,40]. The empirical (relative) nucleophilicity *N* index [41,42] which is based on the HOMO energies obtained within the Kohn–Sham scheme [43] obeys to the expression $N = E_{HOMO}(Nu) - E_{HOMO}(TCE)$, where tetracyanoethylene (TCE) constitutes the reference, as it presents the lowest HOMO energy in a long series of molecules already investigated in the context of polar organic reactions. Besides the global reactivity index, it is possible to define its local (or regional) counterpart condensed to atoms.

The electrophilic $P_k^+$ and nucleophilic $P_k^-$ Parr functions [29] were obtained through the analysis of the Mulliken ASD of the radical anion and the radical cation by single-point energy calculations over the optimized neutral geometries by using the unrestricted UB3LYP formalism for radical species.

### 3.3. X-ray Crystallography

Suitable crystals for X-ray diffraction of **3h** were grown by slow diffusion of *n*-hexane in an ethyl acetate solution. A pale yellow spike of 0.30 × 0.10 × 0.06 mm size, orthorhombic, $P2_12_12_1$, $a = 5.7302(2)$, $b = 14.6518(6)$, $c = 14.7181(6)$ Å, $V = 1235.70(8)$ Å$^3$, $Z = 4$, $D_c = 1.345$ g cm$^{-3}$, was measured at 120(2) K on an Oxford diffraction X caliber diffractometer using graphite-monochromated Mo-Kα radiation ($\lambda = 0.71073$ Å). A ω scan mode was used to collect 18,195 reflections, of which 3042 were independent ($R_{int} = 0.0456$). The crystal structure was solved by direct methods and all non-hydrogen atoms were refined anisotropically on $F^2$ (program SHELXL-2017). The hydrogen atoms for the amino group were located in a difference Fourier synthesis and refined with restrained N–H bond lengths and the other hydrogen atoms were included using a riding model. The crystal absolute structure was not determined. $R1[I > 2\sigma (I)] = 0.0338$, $w$ $R2$(all data) = 0.0582, max. $\Delta\varrho = 0.144$ eÅ$^{-3}$, for 2265 observed reflections and 180 refined parameters. The programs use neutral atom scattering factors, $\Delta f'$ and $\Delta f''$, and absorption coefficients from the *International Tables for Crystallography*.

### 3.4. Protection of Sugars

#### 3.4.1. Preparation of 1,2:3,4-di-O-Isopropylidene-α-D-galactopyranose

*D*-galactose (6 g, 0.034 mol) and 2.5 mL of concentrated sulfuric acid in 300 mL of anhydrous acetone were stirred for 24 h at room temperature. The residual *D*-galactose was filtered on a sintered glass filter. The filtrate was neutralized with a solution of NaHCO$_3$ until the pH = 8, and then filtered to remove the white precipitate of Na$_2$SO$_4$. The acetone was evaporated under reduced pressure and the residue extracted with dichloromethane. The oil obtained, after evaporation of the solvent, was chromatographed on silica gel with hexane/ethyl acetate (4:1 *v/v*) as eluent, yield = 88%, $R_f$ = 0.51 in hexane/ethyl acetate (1:2 v/v). The $^1$H NMR (400 MHz, CDCl$_3$, δ ppm) values were: 1.23, 1.31, 1.37, 1.43 (4s, 12H, 4CH$_3$), 2.30 (1H, OH), 3.70–4.30 (m, 5H, 5CH), 4.60 (dd, 1H, CH, $J$ = 2.40 Hz, $J$ = 2.43 Hz), and 5.61 (d, 1H, CH$_{anomeric}$, $J$ = 2.40 Hz). The $^{13}$C NMR (100MHz, CDCl$_3$, δ ppm) values were: 24.27–25.98 (4CH$_3$), 62.12 (CH$_2$OH), 65.69 (CHO), 68.16 (CHO), 70.68 (CHO), 71.03 (CHO), 96.25 (CHO), and 108.65–109.40 (2Cquaternary).

#### 3.4.2. Preparation of O-Methyl-2,3-O-isopropylidene-D-ribofuranoside

To 20 mL of anhydrous methanol saturated with hydrochloric acid (gaseous) at 0 °C, 0.023 mol of *D*-ribose, 8 mL of dimethoxypropane, and 80 mL of anhydrous acetone were added. The reaction mixture was stirred for 12 h at room temperature. After neutralization with sodium hydroxide and removal of the solvent, the residue was taken up with ether and then washed with water. The pure product was obtained after evaporation of the solvent under reduced pressure, yield: 86%, $R_f$ = 0.65 in hexane/ethyl acetate (2:1 v/v). The $^1$H NMR (400MHz, CDCl$_3$, δ ppm) values were: 1.30 (s, 3H, CH$_3$); 1.47(s, 3H, CH$_3$); 3.45 (s, 3H, OCH$_3$); 3.70 (d, 2H, CH$_2$); 4.60–5.00 (m, 2H, 2CHO); 4.46 (t, 1H, CHO); and 5.06 (s, 1H, H$_{anomeric}$). The $^{13}$C NMR (100MHz, CDCl$_3$, δ ppm) values were: 25.90 (2 CH$_3$), 54.70 (OCH$_3$), 62.30 (CH$_2$OH), 80.40 (CHO), 81.01 (CHO), 82.10 (CHO), 108.40 (CHO), and 111.30 (Cq).

### 3.5. General Procedure for the Tosylation of Sugars

To 0.022 mol of the protected sugar, 7 mL of pyridine was added at 0 °C. The mixture was stirred until it dissolved completely, and then 0.022 mol of tosyl chloride was added in small portions. After 4 h under magnetic stirring at 0 °C, a white solid formed, which was then filtered and subsequently recrystallized from an ethyl acetate/hexane solvent mixture.

### 3.5.1. 3,4-di-O-Isopropylidene-6-O-p-toluenesulfonyl-$\alpha$-D-galactopyranose.

White solid, yield: 75%, $R_f$ = 0.65 in hexane/ethyl acetate (2:1 v/v), MP = 122 °C. The $^1$H NMR (400 MHz, CDCl$_3$, δ ppm) values were: 1.27–1.31–1.34–1.45 (4s, 12H, 4CH$_3$), 2.44 (s, 3H, CH$_{3tosyl}$), 4.05 (m, 2H, CH$_2$OTs), 4.20 (m, 2H, 2CHO), 4.30 (q, 1H, CH, $J$ = 2.5 Hz), 4.56 (dd, 1H, CHO, $J$ = 7.8 Hz, $J$ = 2.4 Hz), 5.45 (d, 1H, CH$_{anomeric}$, $J$ = 5.0 Hz), 7.33 (d, 2H, 2CH$_{aromatic}$, $J$ = 8.0 Hz), and 7.81 (d, 2H, 2CH$_{aromatic}$, $J$ = 8.3Hz). The $^{13}$C NMR (100MHz, CDCl$_3$, δ ppm) values were: 21.36 (CH$_{3tosyl}$), 24.33–24.91–25.8–25.97 (4CH$_3$), 65.85 (CHO), 68.20 (CHO), 70.33 (CHO), 70.39 (CHO), 70.49 (CHO), 96.11 (OCHO), 108.93–109.56 (2Cq), 128.11–129.76 (4C, 4CH$_{aromatic}$), 132.76 (C$_{aromatic}$), and 144.79 (C$_{aromatic}$).

### 3.5.2. O-Methyl-2,3-O-isopropylidene-5-O-*p*-toluenesulfonyl-D-ribofuranoside

White solid, yield: 90%, $R_f$ = 0.9 in hexane/ethyl acetate (2:1 v/v), MP = 84 °C. The $^1$H NMR (400 MHz, CDCl$_3$, δ ppm) values were: 1.28 (s, 3H, CH$_3$), 1.45 (s, 3H, CH$_3$), 2.40 (s, 3H, CH$_{3tosyl}$), 3.33 (s, 3H, OCH$_3$), 3.35 (s, 1H, CHO), 3.95 (d, 2H, CH$_2$), 4.25 (dd, 1H, CHO), 4.5 (m, 1H,CHO), 4.85 (s, 1H, H$_{anomeric}$), and 7.25–7.80 (m, 4H, H$_{aromatic}$). The $^{13}$C NMR (100MHz, CDCl$_3$, δ ppm) values were: 24.30 (CH$_{3tosyl}$), 25.90 (2CH$_3$), 54.70 (OCH$_3$), 65.30 (CH$_2$OTs), 77.40–82.10 (3CHO), 108.40 (CHO), 111.30 (Cq), 129.00–130.60 (4CH$_{aromatic}$), 138.20 (Cq$_{aromatic}$), and 144.30 (Cq$_{aromatic}$).

### 3.6. General Procedure for the Preparation of Sugar Azides

The tosylated sugar (0.005 mol) was mixed with 25 mmol of sodium azide, and 10 mL of dimethylformamide (DMF) was added. The mixture was refluxed for 12 h. After the evaporation of the DMF, the residue obtained was taken up in ethyl acetate to dissolve the product, and then the solution was filtered off to remove the excess of sodium azide. The filtrate was washed with a saturated aqueous solution of NH$_4$Cl, and then twice with water. The organic phase was dried over anhydrous MgSO$_4$, then filtered off and evaporated under a vacuum. The product was purified by chromatography on a column of silica gel by using hexane/ethyl acetate (4:1 *v/v*) as eluent.

### 3.6.1. 6-Azido-1,2: 3,4-di-O-isopropylidene-D–galactopyranose

Yellow pale oil, yield: 84%, $R_f$ = 0.65 in hexane/ethylacetate (2:1 v/v). The $^1$H NMR (400 MHz, CDCl$_3$, δ ppm) values were: 1.32–1.33–1.44–1.53 (4s, 12H, 4CH$_3$), 3.35 (dd, 1H, CH$_2$N$_3$, $J$ = 12.70 Hz, $J$ = 5.30 Hz), 3.50 (dd, 1H, CH$_2$N$_3$, $J$ = 12.70 Hz, $J$ = 7.80 Hz), 3.90 (m, 1H, CH), 4.18 (dd, 1H, CHO, $J$ = 1.90 Hz, $J$ = 7.80 Hz), 4.32 (dd, 1H, CHO, $J$ = 2.49 Hz, $J$ = 5.00 Hz), 4.62 (dd, 1H, CHO, $J$ = 2.48 Hz, $J$ = 7.80 Hz), 5.53 (d, 1H, CH$_{anomeric}$, $J$ = 5.00 Hz). The $^{13}$C NMR (100MHz, CDCl$_3$, δ ppm) values were: 24.38–24.85–25.91–25.99 (4 CH$_3$), 50.62 (CH$_2$), 66.97 (CH), 70.34 (CHO), 70.75 (CHO), 71.13 (CHO), 96.31 (OCHO), and 108.77–109.57 (2Cq).

### 3.6.2. 5-Azido-1-O-methyl-2,3-O-isopropylidene-D-ribofuranoside

Yellow pale oil, yield: 92%, $R_f$ = 0.88 in hexane/ethylacetate (2:1 v/v). The $^1$H NMR (400 MHz, CDCl$_3$, δ ppm) values were: 1.25 (s, 3H, CH$_3$), 1.45 (s, 3H, CH$_3$), 3.20 (td, 1H, CHO, $J$ = 6.81 Hz), 3.30 (s, 3H, OCH$_3$), 3.40 (dd, 1H, CHO), 4.20 (dd, 1H, CHO), 4.50 (s, 2H, CH$_2$N$_3$), and 4.90 (s, 1H, CH$_{anomeric}$). The $^{13}$C NMR (100MHz, CDCl$_3$, δ ppm) values were: 25.90 (2CH$_3$), 53.70 (OCH$_3$), 61.20 (CH$_2$N$_3$), 80.40 (CHO), 80.90 (CHO), 81.70 (CHO), 108.40 (CH$_{anomeric}$), and 111.50 (Cq).

### 3.7. General Procedure for the Synthesis of 1,2,3-Triazoles

Azide (0.751 mmol, 1.2 equivalent), alkyne derivatives (0.622 mmol, 1 equivalent), 0.05 equivalent of copper(I) chloride catalyst, and 0.05 equivalent of poly(pyrazolyl)borate were placed in a reaction tube and 5 mL of water/ethanol (1:1, *v/v*) was added. The mixture was stirred for 24 h at room temperature. After the completion of the reaction as evidenced by TLC, the product was extracted by using diethyl ether. The combined diethyl ether fractions were evaporated under reduced pressure to afford the corresponding final pure 1,2,3-triazole.

### 3.7.1. Synthesis of 1-Benzyl-4-phenyl-1H-1,2,3-triazole (**3a**)

White solid, yield: 94%, $R_f$ = 0.29 in hexane/ethyl acetate (3:1 v/v), MP = 130–132 °C. The $^1$H NMR (400 MHz, CDCl$_3$, δ ppm) values were: 5.60 (s, 2H, CH$_2$), 7.28–7.44 (m, 8H, CH$_{ar}$), 7.68 (s, 1H, CH$_{triazole}$), and 7.81–7.83 (d, 2H, CH$_{ar}$). The $^{13}$C NMR (100MHz, CDCl$_3$, δ ppm) values were: 54.42 (CH$_2$), 119.50 (CH$_{ar}$), 125.89 (3CH$_{ar}$), 127.86 (2CH$_{ar}$), 128.19 (2CH$_{ar}$), 129.18 (CH$_{triazole}$), 130.53 (C$_{ar}$), 134.91 (C$_{ar}$), and 148.25 (C$_{triazole}$). HRMS (FAB+) *m/z*: calculated for C$_{15}$H$_{14}$N$_3$: 236.1188; found: 236.1177.

### 3.7.2. Synthesis of 1-Benzyl-4-(4-fluorophenyl)-1H-1,2,3-triazole (**3b**)

White solid, yield: 93%, $R_f$ = 0.34 in hexane/ethyl acetate (3:1 v/v), MP = 114–116 °C. The $^1$H NMR (400 MHz, CDCl$_3$, δ ppm) values were: 5.59 (s, 2H, CH$_2$), 7.08–7.13 (d, 2H, CH$_{ar}$), 7.28–7.42 (m, 5H, CH$_{ar}$), 7.64 (s, 1H, CH$_{triazole}$), and 7.78–7.80 (d, 2H, CH$_{ar}$). The $^{13}$C NMR (100 MHz, CDCl$_3$, δ ppm) values were: 54.29 (CH$_2$), 115.69 (2CH$_{ar}$), 115.91 (2CH$_{ar}$), 119.24 (CH$_{ar}$), 127.41 (C$_{ar}$), 127.49 (2CH$_{ar}$), 128.11 (CH$_{triazole}$), 128.86 (C$_{ar}$), 129.21 (C$_{triazole}$), and 134.59 (CF$_{ar}$). HRMS (FAB+) *m/z*: calculated for C$_{15}$H$_{13}$N$_3$F: 254.1094; found: 254.1087.

### 3.7.3. Synthesis of 1-Benzyl-4-p-tolyl-1H-1,2,3-triazole (**3c**)

White solid, yield: 89%, $R_f$ = 0.38 in hexane/ethyl acetate (3:1 v/v), MP = 152–154 °C. The $^1$H NMR (400 MHz, CDCl$_3$, δ ppm) values were: 2.38 (s, 3H, CH$_3$), 5.56 (s, 2H, CH$_2$), 7.21–7.40 (m, 7H, CH$_{ar}$), 7.52 (s, 1H, CH$_{triazole}$), 7.58 (d, 2H, CH$_{ar}$). The $^{13}$C NMR (100 MHz, CDCl$_3$, δ ppm) values were: 21.29 (CH$_3$), 54.18 (CH$_2$), 119.23 (CH$_{ar}$), 125.62 (2CH$_{ar}$), 127.74 (2CH$_{ar}$), 128.07 (2CH$_{ar}$), 128.75 (2CH$_{ar}$), 129.14 (C$_{ar}$), 129.50 (CH$_{triazole}$), 134.78 (C$_{ar}$), 138.02 (C$_{ar}$), and 148.30 (C$_{triazole}$). HRMS (FAB+) *m/z*: calculated for C$_{15}$H$_{14}$N$_3$: 250.1344; found: 250.1327.

### 3.7.4. Synthesis of Methyl 4-(1-Benzyl-1H-1,2,3-triazol-4-yl)benzoate (**3d**)

Yellow solid, yield: 76%, $R_f$ = 0.22 in hexane/ethyl acetate (3:1 *v/v*). MP = 164–166 °C. The $^1$H NMR (400 MHz, CDCl$_3$, δ ppm) values were: 3.94 (s, 3H, CH$_3$), 5.61 (s, 2H, CH$_2$), 7.28–7.40 (m, 5H, CH$_{ar}$), 7.76 (s, 1H, CH$_{triazole}$), 7.88–7.90 (d, 2H, CH$_{ar}$); 8.07–8.10 (d, 2H, CH$_{ar}$). The $^{13}$C NMR (100 MHz, CDCl$_3$, δ ppm) values were: 52.15 (CH$_3$), 54.38(CH$_2$), 120.37 (CH$_{ar}$), 125.48 (2CH$_{ar}$), 128.15 (2CH$_{Ar}$), 128.95 (2CH$_{ar}$), 129.25 (2CH$_{ar}$), 129.61 (C$_{ar}$), 130.20 (CH$_{triazole}$), 134.41 (C$_{ar}$), 134.82 (C$_{ar}$), 147.21 (C$_{triazole}$), and 166.79 (C$_{carbonyl}$). HRMS (FAB+) *m/z*: calculated for C$_{17}$H$_{16}$N$_3$O$_2$: 294.1243; found: 294.1243.

### 3.7.5. Synthesis of 4-(1-Benzyl-1H-1,2,3-triazol-4-yl)benzaldehyde (**3e**)

White solid, yield: 93%, $R_f$ = 0.14 in hexane/ethyl acetate (3:1 v/v), MP = 134–135 °C. The $^1$H NMR (400 MHz, CDCl$_3$, δ ppm) values were: 5.62 (s, 2H, CH$_2$), 7.28–7.43(m, 5H, CH$_{ar}$), 7.80 (s, 1H, CH$_{triazole}$), 7.92–7.94 (d, 2H, CH$_{ar}$), 7.98–8.00 (d, 2H, CH$_{ar}$), and 10.02 (s, 1H, CHO). The $^{13}$C NMR (100 MHz, CDCl$_3$, δ ppm) values were: 54.42 (CH$_2$), 120.70 (CH$_{ar}$), 126.04 (2CH$_{ar}$), 128.18 (2CH$_{ar}$), 129.01 (2CH$_{ar}$), 129.28 (2CH$_{Ar}$), 130.38 (CH$_{triazole}$), 134.32 (CH$_{ar}$), 135.80 (CH$_{ar}$), 136.31 (CH$_{ar}$), 146.93 (C$_{triazole}$), and 191.72 (CHO). HRMS (FAB+) *m/z*: calculated for C$_{16}$H$_{14}$N$_3$O: 264.1137; found: 294.1134.

### 3.7.6. Synthesis of 1-Benzyl-4-(4-phenoxyphenyl)-1H-1,2,3-triazole (**3f**)

Yellow solid, yield: 91%, $R_f$ = 0.3 in hexane/ethyl acetate (3:1 v/v), MP = 173–175 °C. The $^1$H NMR (400 MHz, CDCl$_3$, δ ppm) values were: 5.59 (s, 2H, CH$_2$), 7.05–7.15 (m, 5H, CH$_{ar}$), 7.28–7.42 (m, 7H, CH$_{ar}$), 7.64 (s, 1H, CH$_{triazole}$), and 7.77–7.79 (d, 2H, CH$_{ar}$). The $^{13}$C NMR (100 MHz, CDCl$_3$, δ ppm) values were: 54.25 (CH$_2$), 119.04 (2CH$_{ar}$), 119.08 (2CH$_{ar}$), 123.49 (CH$_{ar}$), 125.70 (CH$_{ar}$), 127.22 (C$_{ar}$), 128.08 (2CH$_{ar}$), 128.81 (2CH$_{ar}$), 129.18 (2CH$_{ar}$), 129.82 (2CH$_{Ar}$), 130.10 (CH$_{triazole}$), 134.70 (C$_{ar}$), 156.95 (C$_{triazole}$), and 157.33(2C$_{ar}$). HRMS (FAB+) *m/z*: calculated for C$_{21}$H$_{18}$N$_3$O: 328.1449; found: 328.1449.

### 3.7.7. Synthesis of 1-Phenethyl-4-phenyl-1H-1,2,3-triazole (**3g**)

White solid, yield: 88%, $R_f$ = 0.55 in hexane/ethyl acetate (3:2 v/v), MP = 142 °C. The $^1$H NMR (400 MHz, CDCl$_3$, δ ppm) values were: 3.23–3.28 (t, 2H, CH$_2$), 4.61–4.66 (t, 2H, CH$_2$), 7.12–7.15 (d, 1H,

CH$_{ar}$), 7.26–7.42 (m, 7H, CH$_{ar}$), 7.46 (s, 1H, CH$_{triazole}$), and 7.75–7.78 (d, 2H, CH$_{ar}$). The $^{13}$C NMR (100 MHz, CDCl$_3$, δ ppm) values were: 37.20 (CH$_2$), 52.15 (CH$_2$), 120.26 (CH$_{ar}$), 121.53 (2CH$_{ar}$), 126.09 (2CH$_{ar}$), 127.55 (2CH$_{ar}$), 128.47 (CH$_{ar}$), 129.14 (2CH$_{ar}$), 129.20 (CH$_{triazole}$), 129.26 (C$_{ar}$), 130.91 (C$_{ar}$), and 137.47 (C$_{triazole}$). HRMS (FAB+) *m/z:* calculated for C$_{16}$H$_{16}$N$_3$: 250.1344; found: 250.1348.

### 3.7.8. Synthesis of 4-(1-Benzyl-1H-1,2,3-triazol-4-yl)benzenamine (**3h**)

White solid, yield: 73%, R$_f$ = 0.48 in hexane/ethyl acetate (1:2 v/v), MP = 184 °C. The $^1$H NMR (400 MHz, CDCl$_3$, δ ppm) values were: 4.85 (s, 2H, NH$_2$), 5.56 (s, 2H, CH$_2$), 6.72–6.75 (d, 2H, CH$_{ar}$), 7.31–7.35 (m, 5H, CH$_{ar}$), 7.49–7.52 (d, 2H, CH$_{ar}$), and 8.05 (s, 1H, CH$_{triazole}$). The $^{13}$C NMR (100 MHz, CDCl$_3$, δ ppm) values were: 55.36 (CH$_2$), 116.83 (2CH$_{ar}$), 121.07 (C$_{ar}$), 121.44 (CH$_{ar}$), 128.17 (2CH$_{ar}$), 129.43 (2CH$_{ar}$), 129.96 (2CH$_{ar}$), 130.44 (CH$_{triazole}$), 137.32 (Car), 149.90 (C$_{triazole}$), and 150.36 (C$_{ar}$). HRMS (FAB+) m/z: calculated for C$_{15}$H$_{15}$N$_4$: 251.1297; found: 251.1299.

### 3.7.9. Synthesis of 1-Benzyl-4-(((2,2,7,7-tetramethyltetrahydro-5H-bis([1,3]dioxolo)[4,5-b:4′,5′-d]pyran-5-yl)methoxy)methyl)-1H-1,2,3-triazole (**3i**)

White solid yield: 83%, *R$_f$* = 0.8 in hexane/ethyl acetate (1:2 v/v), MP = 95 °C. The $^1$H NMR (400 MHz, CDCl$_3$, δ ppm) values were: 1.32 (s, 6H, CH$_3$), 1.42 (s, 3H, CH$_3$), 1.52 (s, 3H, CH$_3$), 3.68–3.71 (d, 2H, CH$_2$), 3.99 (s, 2H, CH$_2$), 4.24–4.32 (d, 2H, CH), 4.58–4.70 (d, 3H, CH), 5.51 (s; 2H, CH$_2$), 7.27–7.36 (m, 5H, CH$_{ar}$), 7.52 (s, 1H, CH$_{triazole}$). The $^{13}$C NMR (100 MHz, CDCl$_3$, δ ppm) values were: 24.30 (CH$_3$), 24.91 (CH$_3$), 25.97 (CH$_3$), 26.06 (CH$_3$), 54.19 (CH$_2$), 58.59 (CH$_2$), 64.91 (CH$_2$), 66.67 (CH), 68.72 (CH), 69.38 (CH), 70.57 (CH), 71.38 (CH), 96.33 (2C), 108.56 (CH$_{triazole}$), 109.25 (CH$_{ar}$), 128.15 (2CH$_{ar}$), 128.60 (2CH$_{ar}$), 128.99 (C$_{ar}$), and 134.49 (C$_{triazole}$).

### 3.7.10. Synthesis of 4-Phenyl-1-((2,2,7,7-tetramethyltetrahydro-5H-bis([1,3]dioxolo)[4,5-b:4′,5′-d]pyran-5-yl)methyl)-1H-1,2,3-triazole (**3j**)

White solid, yield: 89%, *R$_f$* = 0.5 in hexane/ethyl acetate (3:2 v/v), MP = 137 °C. The $^1$H NMR (400 MHz, CDCl$_3$, δ ppm) values were: 1.22 (s, 3H, CH$_3$), 1.30 (s, 3H, CH$_3$), 1.33 (s, 3H, CH$_3$), 1.44 (s, 3H, CH$_3$), 4.15–4.17 (d, 2H, CH$_2$), 4.176–4.27 (q, H, CH), 4.33–4.38 (t, H, CH), 4.56–4.6 (d, 2H, CH), 5.46–5.48 (d, H, CH), 7.25–7.37 (m, 3H, CH$_{ar}$), 7.76–7.78 (d, 2H, CH$_{ar}$), and 7.90 (s, 1H, CH$_{triazole}$). The $^{13}$C NMR (100 MHz, CDCl$_3$, δ ppm) values were: 24.82 (CH$_3$), 25.29 (CH$_3$), 26.34 (CH$_3$), 30.109 (CH$_3$), 51.01 (CH$_2$), 67.67 (CH), 70.72 (CH), 71.00 (CH), 71.14 (CH), 71.60 (CH), 96.66 (2C), 109.52 (2CH$_{ar}$), 110.28 (CH$_{ar}$), 126.13 (2CH$_{ar}$), 128.42 (CH$_{triazole}$), 129.19 (C$_{ar}$), and 131.18 (C$_{triazole}$).

### 3.7.11. Synthesis of 1-((6-Methoxy-2,2-dimethyltetrahydrofuro [3,4-d][1,3]dioxol-4-yl)methyl)-4-(((2,2,7,7-tetramethyltetrahydro-5H-bis([1,3]dioxolo)[4,5-b:4′,5′-d]pyran-5-yl)methoxy)methyl)-1H-1,2,3-triazole (**3k**)

White solid, yield: 80%, *R$_f$* = 0.62 in hexane/ethyl acetate (1:1 v/v). The $^1$H NMR (400 MHz, CDCl$_3$, δ ppm) values were: 1.22–1,51 (m, 18H, CH$_3$), 3.34 (s, 3H, OCH$_3$), 3.63–3.68 (m, 1H, CH$_2$), 3.93–3.96 (t, H,CH$_2$), 4.16–4.27 (m, 2H, CH$_2$), 4.32–4.40 (q, 1H, CH), 4.50–4.63 (m, 4H, CH), 4.67–4.71 (m, 2H, CH), 4.96 (s, 2H, CH$_2$), 5.26 (s, 1H, CH), 5.48–5.49 (d, 1H, CH), and 7.65 (s, 1H, CH$_{triazole}$). The $^{13}$C NMR (100 MHz, CDCl$_3$, δ ppm) values were: 24.28 (CH$_3$), 24.88 (CH$_3$), 26.06 (CH$_2$), 26.40 (CH$_3$), 27.04 (CH$_3$), 29.56 (CH$_3$), 35.06 (CH$_2$), 55.53 (OCH$_3$), 64.82 (CH$_2$), 66.72 (CH$_2$), 68.6(CH), 69.45 (CH), 70.46 (CH), 70.72 (CH), 71.12 (CH), 81.76 (CH), 84.96 (CH), 95.16 (CH), 96.31 (C), 108.53 (CH), 109.29 (C), 109.44 (C), 112.82 (CH$_{triazole}$), and 122.95 (C$_{triazole}$).

### 3.7.12. Synthesis of 4-Phenyl-1-(12-(4-phenyl-1H-1,2,3-triazol-1-yl)dodecyl)-1H-1,2,3-triazole (**3l**)

White solid, yield: 90%, R$_f$ = 0.57 in hexane/ethyl acetate (3:2 v/v), MP = 177 °C. The $^1$H NMR (400 MHz, CDCl$_3$, δ ppm) values were: 1.27 (s, 16H, CH$_2$); 1.93–1,98 (q, 4H, CH$_2$), 4.41 (t, 4H, CH$_2$), 7.22–7.53 (m, 6H, CH$_{ar}$), 7.75 (s, 1H, CH$_{triazole}$), and 7.85 (d, 2H, CH$_{ar}$). The $^{13}$C NMR (100 MHz, CDCl$_3$, δ ppm) values were: 26.84 (2CH$_2$), 29.33 (2CH$_2$), 29.67 (2CH$_2$), 3.72 (4CH$_2$), 50.83 (2CH$_2$), 119.75 (4CH$_{ar}$),

122.00 (2CH$_{ar}$), 126.08 (4CH$_{ar}$), 128.48 (2CH$_{triazole}$), 129.22 (2C$_{Ar}$), and 139.00 (C$_{triazole}$). HRMS (FAB+) *m/z*: calculated for C$_{28}$H$_{37}$N$_6$: 457.308; found: 457.3088.

3.7.13. Synthesis of 1,12-bis(4-(((2,2,7,7-Tetramethyltetrahydro-5H-bis([1,3]dioxolo)[4,5-b:4',5'-d]pyran-5-yl)methoxy)methyl)-1H-1,2,3-triazol-1-yl)dodecane (**3m**)

White solid, yield: 80%, $R_f$ = 0.4 in hexane/ethyl acetate (1:2 v/v). The $^1$H NMR (400 MHz, CDCl$_3$, δ ppm) values were: 1.21–1.50 (m, 4H, CH$_2$ + CH$_3$), 1.84–1.87 (m, 4H, CH$_2$), 3.64–3.69 (m, 4H, CH$_2$O), 3.96–3.98 (t, 4H, CH$_2$), 4.27–4.31 (m, 4H, CH), 4.55–4.58 (q, 2H, CH), 4.67–4.68 (d, 2H, CH), 5.27 (s, 4H, CH$_2$O), 5.50–5.51 (d, 2H, CH), and 7.54 (s, 1H, CH$_{triazole}$). The $^{13}$C NMR (100 MHz, CDCl$_3$, δ ppm) values were: 24.44 (4CH$_3$), 24.89 (CH$_3$), 25.96 (CH$_3$), 26.04 (CH$_3$), 26.43 (CH$_3$), 26.93 (4CH$_2$), 29.27 (2CH$_2$), 29.36 (2CH$_2$), 30.24 (2CH$_2$), 50.24 (2CH$_2$), 64.83 (2OCH$_2$), 66.71 (2CHO), 69.25 (2CH), 70.47 (2CH), 70.67 (2CH), 71.13 (2CH), 96.31 (2CH), 108.54 (2C), 109.23 (2C), 122.35 (2CH$_{triazole}$), and 145.06 (2C$_{triazole}$). HRMS (FAB+) *m/z*: calculated for C$_{42}$H$_{68}$N$_6$Na: 871.4793; found: 871.4791.

3.7.14. Synthesis of 1,12-bis(4-(((2,2,7,7-Tetramethyltetrahydro-5H-bis([1,3]dioxolo)[4,5-b:4',5'-d]pyran-5-yl)oxy)methyl)-1H-1,2,3-triazol-1-yl)dodecane (**3n**)

White solid, yield: 80%, $R_f$ = 0.37 in hexane/ethyl acetate (1:2 v/v). The $^1$H NMR (400 MHz, CDCl$_3$, δ ppm) values were: 1.20–1.50 (m, 4H, CH$_2$ + CH$_3$), 1.84–1.87 (m, 2H, CH$_2$), 3.39–4.07 (m, 4H, CH$_2$), 4.25–4.31 (dd, 2H, CH); 4.74–4.77 (d, 2H, CH), 4.71–4.80 (d, 2H, CH), 5.26 (s, 2H, CH), 5.82 (s, 2H, CH), and 7.56 (s, 1H, CH$_{triazole}$). The $^{13}$C NMR (100 MHz, CDCl$_3$, δ ppm) values were: 25.56 (4CH$_3$), 26.18 (4CH$_3$), 26.84 (2CH$_2$), 28.93 (2CH$_2$), 29.37 (2CH$_2$), 30.26 (6CH$_2$), 50.35 (2CH$_2$), 64.19 (2OCH$_2$), 67.35 (2 CH), 72.38 (4CH), 81.06 (2CH), 81.63 (2CH), 82.67 (C), 106.21 (C), 109.10 (C), 111.82 (C), 122.34 (2CH$_{triazole}$), and 144.74 (2C$_{triazole}$).

## 4. Conclusions

We have demonstrated that homoscorpionate poly(pyrazolyl)borate ligands such as HB(pz)$_3^-$ and H$_2$B(pz)$_2^-$ anions can act as remarkable in situ stabilizing ligands of copper(I) catalysts. The resulting poly(pyrazolyl)borate-copper(I) catalyzes effectively and regioselectively the cycloaddition of azides with alkynes in water/ethanol as solvent mixture under ambient conditions, affording 1,4-disubstituted 1,2,3-triazole derivatives. A variety of functional groups were found to be compatible with this process, including the crowded and complex carbohydrate substrates. Analysis of the global and local reactivity defined within the CDFT shows that poly(pyrazolyl)borate-copper(I) greatly enhances the polar character of the azide-alkyne [3+2] cycloaddition, by generating a very strong electrophilic [dihydrobis(pyrazolyl)borate]copper(I)-acetilyde intermediate. These results further illustrate the power of the synergy between experimental and computational methods in catalysis.

**Supplementary Materials:** The following are available online at www.mdpi.com/xxx/s1.

**Author Contributions:** Concept and strategy implementation were set up by S.-E.S. and H.B.; synthetic work was performed by B.L. and H. B.; computational work was executed and supervised by L.R.D., L.B., and H.B.; data analysis and interpretation of the catalytic and computational results were carried out by L.B., H.A., L.R. D.M.J., and S.-E.S. Finally, the writing of the manuscript was done by H.B., B.L., L.R.D., M.J., and S.-E.S.

**Acknowledgments:** Financial support from Université Cadi Ayyad and the Spanish Ministerio de Ciencia e Innovación (MCINN) (Projects CTQ2016-75068P and Unidad de Excelencia María de Maetzu MD2015-0538) is gratefully acknowledged.

**Conflicts of Interest:** The authors declare no conflict of interest.

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
