# Peer review of "Clicking Azides and Alkynes with Poly(pyrazolyl)borate-Copper(I) Catalysts: An Experimental and Computational Study"

_catalysts, doi:10.3390/catal9080687_

Round 1

Reviewer 1 Report

This paper describes the development of a Click reaction between azides and alkynes, including examples bearing sugar substituents, catalysed by Cu(I) HB(pz)3- or Cu(I) H2B(pz)2-under mild reaction conditions to provide 1,4-disubstituted-1,2,3-triazoles.

There are a number of questions that I hope the authors could address:

1)      The authors apparently did DFT calculations to predict the structure of Cu(I) HB(pz)3- and Cu(I) H2B(pz)2-. Other such poly(pyrazolyl)borate-copper complexes have been reported albeit the Cu oxidation state being +2 (see J. Coord. Chem. 2007, 60, 163, DOI: 10.1080/00958970600793828). Naturally one would ask what is the true catalytic species given that no X-ray crystal structure is available to confirm that if both Cu(I) HB(pz)3- and Cu(I) H2B(pz)2- do really exist and is not the oxidized Cu(II) form that is being generated? The authors if unable to crystallise the complexes can run mass spec (e.g. ESI) to confirm the species.

2)      Each catalyst used in the alkyne-azide Click reaction was tested with various solvents or solvent combinations. However I do not agree with the statement “Indeed, various solvents were examined (Table 2) pointing out that the solvent plays a significant role concerning the obtained yield” on page 4, lines 119-120, as the reported yields for all reactions using the same catalytic system are within 6% difference of one another. They seem to be working well for most polar solvent systems. The authors should try other less polar solvents like octane, toluene or DCM used in screening in Table 4 with the borate Cu(I).

3)      The reaction was then extended to a variety of alkynes bearing aryl/sugar substituents and azides bearing alkyl/benzyl/sugar substituents with 70-93% yield. The experimental section concludes with a comparison of the developed methodology to other reported Cu(I) catalytic systems. However I do not agree with the statement “Our catalytic systems benefit from their easiest synthetic protocol and from their higher activity in a mixture of environmentally benign solvents such as ethanol and water.” on page 7, lines 170-171. With regards to the synthetic protocol, there needs to be further description of the difficulties in use of the previously reported synthetic protocols to justify the claim that it is “easiest”. With regards to the higher activity in environmentally benign solvents, it appears from the table that the Cu(I)-TBTM system in fact outperforms the developed protocol in terms of yield, catalyst loading and reaction time? I suggest the authors remove the statement and instead mention that the developed protocol is comparable to previously reported methods.

4)      DFT calculations were used to understand the high regioselectivity of the reaction by computing the electrophilicity and nucleophilicity of each component of the Click reaction before and after coordination of the copper catalyst. Please include reference(s) for steps described on page 9, lines 237-242. I find that using CDFT descriptors only give qualitative predictions. The authors should further confirm this selectivity by calculating the pathways for intermediates and transition states leading to various regioisomers for the 3+2 cycloaddition reaction. Comparison of the energetics of the pathway s leading to the regioisomers would be much more insightful in combination with CDFT.

5)      Consistency in reporting of NMR data (some are reported with 1 decimal point, others with 2 decimal points).

6)      Some compounds lack mass spec data.

Author Response

We would like to thank the two reviewers for appreciating our research work and for their positive reports on our manuscript (catalysts-554244).

Please find hereunder our “point-by-point” answers to the inquiries raised by them. The new additions and corrections are given in red color in the mean text of the manuscript.

Reviewer 1:

This paper describes the development of a Click reaction between azides and alkynes, including examples bearing sugar substituents, catalysed by Cu(I) HB(pz)3- or Cu(I) H2B(pz)2-under mild reaction conditions to provide 1,4-disubstituted-1,2,3-triazoles.

There are a number of questions that I hope the authors could address:

1)      The authors apparently did DFT calculations to predict the structure of Cu(I) HB(pz)3- and Cu(I) H2B(pz)2-. Other such poly(pyrazolyl)borate-copper complexes have been reported albeit the Cu oxidation state being +2 (see J. Coord. Chem. 2007, 60, 163, DOI: 10.1080/00958970600793828). Naturally one would ask what is the true catalytic species given that no X-ray crystal structure is available to confirm that if both Cu(I) HB(pz)3- and Cu(I) H2B(pz)2- do really exist and is not the oxidized Cu(II) form that is being generated? The authors if unable to crystallise the complexes can run mass spec (e.g. ESI) to confirm the species.

Authors: A major problem in using copper(I), which is the catalytically active species in CuAAC, is its thermodynamic instability and easy oxidation to copper(II). In order to circumvent this, poly(pyrazolyl)borate ligands were reported as excellent stabilizing ligands for copper(I) leading to very stable Cu(I) complexes (See references [11-14] and [16]). In addition, the presence of terminal alkynes leads to the reduction of copper(II) to copper(I), thus maintaining the generation and presence of copper(I) in the catalytic medium, under the known Glaser reaction ( J. Am. Chem. Soc. 2014, 136, 16760 & J. Am. Chem. Soc. 2014, 136, 924).

We agree with the reviewer that Electrospray Ionisation Mass spectrometry is an excellent tool to detect intermediate species in CuAAC, an issue which is under investigation in collaboration with experts in this spectrometry method.

2)      Each catalyst used in the alkyne-azide Click reaction was tested with various solvents or solvent combinations. However I do not agree with the statement “Indeed, various solvents were examined (Table 2) pointing out that the solvent plays a significant role concerning the obtained yield” on page 4, lines 119-120, as the reported yields for all reactions using the same catalytic system are within 6% difference of one another. They seem to be working well for most polar solvent systems. The authors should try other less polar solvents like octane, toluene or DCM used in screening in Table 4 with the borate Cu(I).

AuthorsWe thank the reviewer’s suggestion that points to the use of non polar organic solvents in CuAAC. However, our main objective is to perform the CuAAC under strict click conditions, in order to set up a greener chemical protocol for CuAAC. To do so, we perform our catalytic study in environmentally benign solvents such as water, ethanol and their mixtures.

3)      The reaction was then extended to a variety of alkynes bearing aryl/sugar substituents and azides bearing alkyl/benzyl/sugar substituents with 70-93% yield. The experimental section concludes with a comparison of the developed methodology to other reported Cu(I) catalytic systems. However I do not agree with the statement “Our catalytic systems benefit from their easiest synthetic protocol and from their higher activity in a mixture of environmentally benign solvents such as ethanol and water.” on page 7, lines 170-171. With regards to the synthetic protocol, there needs to be further description of the difficulties in use of the previously reported synthetic protocols to justify the claim that it is “easiest”. With regards to the higher activity in environmentally benign solvents, it appears from the table that the Cu(I)-TBTM system in fact outperforms the developed protocol in terms of yield, catalyst loading and reaction time? I suggest the authors remove the statement and instead mention that the developed protocol is comparable to previously reported methods.

Authors: Following the reviewer’s suggestion, the statement isnow appears as "Our catalytic systems show higher activity in CuAAC by using a mixture of environmentally benign solvents such as ethanol and water, in a similar performanceto those previously reported, in particular to the well studied Cu(I)-TBTM system." (page 7)

4)   DFT calculations were used to understand the high regioselectivity of the reaction by computing the electrophilicity and nucleophilicity of each component of the Click reaction before and after coordination of the copper catalyst. Please include reference(s) for steps described on page 9, lines 237-242. I find that using CDFT descriptors only give qualitative predictions. The authors should further confirm this selectivity by calculating the pathways for intermediates and transition states leading to various regioisomers for the 3+2 cycloaddition reaction. Comparison of the energetics of the pathways leading to the regioisomers would be much more insightful in combination with CDFT. 

Authors: Two more new references describing the key intermediates in the CuAAC mechanism are now added as reference [33] and [34].In addition, the regioselectivity in CuAAC was systematically studied, using the DFT methodology, by Calvo-Losada et al. as well as by some of us as reported in the references [35] and [36].

5)  Consistency in reporting of NMR data (some are reported with 1 decimal point, others with 2 decimal points).

Authors:  Allvalues reported in NMR dataarenow presented with two decimal points.

6)   Some compounds lack mass spec data.

Authors: 1H,13C NMR as well as melting points data were found to be enough to elucidate the final structures of the compounds lacking of HRMS data.

Reviewer 2 Report

Stiriba et al. report here on the use of scorpionate ligands (namely tris- and bis- pyrazolylborates) to stabilize the Cu(I) in situ to catalyze the [2+3] cycloaddition also known as azide click reaction. The present results are compared with the literature data on already existing catalysts, and the reactivity is discussed and supported by theoretical calculations.

The discussed results are somewhat expected and not so revolutionizing, but the authors also report some new crystal structures together with some novel reactions.

Overall, this work deserves to be published in the Catalyst after the comments presented below are fully addressed.

1) The literature referencing could be improved as these ligands have been widely used.

2) The catalytic cycle proposed in the paper: a) Cannot be deduced from existing experimental data in this work; b) Is already known and is published (see for instance ref Worell, J. A. Malik, V. V. Fokin, Science, 2013, 340, 457-460), however the authors do not site this milestone work.

3) The control experiments are missing: a) One can see that the product yields when the bidentate ligand was used were somewhat lower. It can be easily explained by the partial oxidation of Cu(I) to Cu(II) as the bidentate ligand will preferentially form Cu(L)2 complexes with tetrahedral coordination of copper. What if the authors add the catalyst with the 1:2 ratio of metal to ligand?

4) What is the reproducibility of reported yields?

5) The name of the paper is overly pretentious. When talking about the synergy one implies some emerging phenomena. Instead here, by using the word synergy, the authors mean that the theory supports experimental data which is too much pretentious and can mislead the reader. Hence, it should be modified correspondingly.

Author Response

We would like to thank the two reviewers for appreciating our research work and for their positive reports on our manuscript (catalysts-554244).

Please find hereunder our “point-by-point” answers to the inquiries raised by them. The new additions and corrections are given in red color in the mean text of the manuscript.

Reviewer 2 :

Stiriba et al. report here on the use of scorpionate ligands (namely tris- and bis- pyrazolylborates) to stabilize the Cu(I) in situ to catalyze the [2+3] cycloaddition also known as azide click reaction. The present results are compared with the literature data on already existing catalysts, and the reactivity is discussed and supported by theoretical calculations.

The discussed results are somewhat expected and not so revolutionizing, but the authors also report some new crystal structures together with some novel reactions.

Overall, this work deserves to be published in the Catalyst after the comments presented below are fully addressed.

1) The literature referencing could be improved as these ligands have been widely used.

Authors: Following the reviewer’s suggestions, additional references [11-17] were included in the introduction section.

2) The catalytic cycle proposed in the paper: a) Cannot be deduced from existing experimental data in this work; b) Is already known and is published (see for instance ref Worell, J. A. Malik, V. V. Fokin, Science, 2013, 340, 457-460), however the authors do not site this milestone work.

Authors: Two references that reported findings on thekey intermediates in the CuAAC mechanism are now added in the paragraph describing the proposed catalytic cycle and appeared as references [33] and [34].

3) The control experiments are missing: a) One can see that the product yields when the bidentate ligand was used were somewhat lower. It can be easily explained by the partial oxidation of Cu(I) to Cu(II) as the bidentate ligand will preferentially form Cu(L)2 complexes with tetrahedral coordination of copper. What if the authors add the catalyst with the 1:2 ratio of metal to ligand?

Authors: Control experiments with only the copper(I) salt were investigated and the results are shown in Table 4 and are described on page7.

4) What is the reproducibility of reported yields?

Authors: We thank the reviewer for his/her suggestion. Indeed, we checkthe reproducibility of our catalyst and the results show that the yields are almost the same in all reproducibility tests.

5) The name of the paper is overly pretentious. When talking about the synergy one implies some emerging phenomena. Instead here, by using the word synergy, the authors mean that the theory supports experimental data which is too much pretentious and can mislead the reader. Hence, it should be modified correspondingly.

Authors: Following the reviewer’s suggestion, the title of the paper was modified and now appears as "Clicking Azides and Alkynes with Poly(pyrazolyl)borate-Copper(I) Catalysts: An Experimental and Computational study"

Round 2

Reviewer 1 Report

No further comments.

Reviewer 2 Report

The authors responded to the majority of comments.

The paper was improved correspondingly.

Some minor corrections are still necessary:

1) Figure 1 is missing from the PDF; it might be the problem of conversion, worth checking.

Otherwise, I suggest the present paper for publication.